# Short- and Long-Term Effects of Inhaled Ultrafine Particles on Blood Pressure: A Systematic Review and Meta-Analysis

**DOI:** 10.3390/jcm13226802

**Published:** 2024-11-12

**Authors:** Joanna Izabela Lachowicz, Paweł Gać

**Affiliations:** Department of Population Health, Division of Environmental Health, Occupational Medicine and Epidemiology, Wroclaw Medical University, Mikulicza-Radeckiego 7, PL 50-368 Wroclaw, Poland; pawel.gac@umw.edu.pl

**Keywords:** ultrafine particles (UFP), systolic blood pressure, diastolic blood pressure, meta-analysis, golden ratio

## Abstract

**Background:** Air pollution contributes to up to 60% of premature mortality worldwide by worsening cardiovascular conditions. Ultrafine particles (UFPs) may negatively affect cardiovascular outcomes, and epidemiological studies have linked them to short- and long-term blood pressure (BP) imbalance. **Methods:** We conducted a systematic review and meta-analysis of the short- and long-term effects of UFP exposure on systolic (SBP) and diastolic (DBP) blood pressure. Eligibility criteria were established using the Population, Exposure, Comparator, Outcome, and Study Design (PECOS) model, and literature searches were conducted in Web of Science, PubMed, Embase, and Scopus for studies published between 1 January 2013 and 9 October 2024. Risk of Bias (RoB) was assessed following World Health Organization (WHO) instructions. Separate meta-analyses were performed for the short- and long-term effects of UFP exposure on SBP and DBP. Additionally, we analyzed SBP and DBP imbalances across different timespans following short-term exposure. **Results:** The results showed an increase in BP during short-term UFP exposure, which returned to baseline values after a few hours. Changes in SBP were greater than in DBP following both short- and long-term exposure. Prolonged exposure to UFPs is associated with increased SBP and concurrently low DBP values. Chronic exposure to UFPs may lead to a persistent increase in SBP, even without a concurrent increase in DBP. **Conclusions:** The findings presented here highlight that UFPs may contribute to worsening cardiovascular outcomes in vulnerable populations living in air-polluted areas.

## 1. Introduction

Ambient outdoor air pollution was estimated to cause 4.2 million premature deaths worldwide in 2019 (WHO, Ambient (outdoor) air quality and health). Lifelong exposure to air pollutants can shorten life expectancy by up to 6 months in moderately polluted countries and by 1–2 years in highly polluted areas [1]. Air pollution is associated with the worsening of cardiovascular conditions, such as coronary artery disease [2], venous thromboembolism, and cardiac arrhythmia and arrest, and accounts for up to 60% of premature mortality worldwide [2]. The World Health Organization (WHO) estimated that in 2019, approximately 37% of outdoor air pollution-related premature deaths were due to ischemic heart disease and stroke [3].

Air pollutants are generated in natural events (e.g., forest fires). However, most chemical and physical agents that alter atmospheric characteristics are emitted by household combustion tools, motor vehicles, and industrial equipment. Pollutants of major public health concern include carbon monoxide (CO), ozone (O_3_), nitrogen oxides (NO_x_), sulfur dioxide (SO_2_), and particulate matter (PM) [3]. PM is a heterogeneous combination of particles with varying sizes. Typically, PM refers to respirable size fractions ranging from 1 to 10 μm in aerodynamic diameter, in contrast to UFPs, which have diameters below 100 nm. Compared to other air pollutants, UFPs are suggested to have unique negative health effects due to their high pulmonary deposition efficiency, increased translocation into circulation, and significant surface area-to-mass ratio [4,5].

It is well-established that exposure to PM affects the cardiovascular system acutely (e.g., increased BP; alterations in heart rate variability (HRV)) or chronically (e.g., exacerbation of atherosclerosis) [6]. However, less is known about the effects of UFPs on cardiovascular risks.

Studies conducted over the last 20 years have shown both short- and long-term effects of UFP exposure. Short-term effects (<7 days) include respiratory symptoms, pulmonary and systemic inflammation, HRV changes, and BP alterations, while long-term effects (>7 days) are associated with increased mortality (all-cause, cardiovascular, and pulmonary) and various types of morbidity [3].

Even though there is growing evidence of the role of UFPs in worsening cardiovascular conditions in populations living in air-polluted areas, relatively few epidemiological studies assess the link between UFP exposure and cardiovascular diseases (CVDs). Moreover, comparison among the few available results is complicated due to the variety of experimental protocols, inconsistencies in UFP size distribution, exposure duration, and statistical models and calculations.

Over the last decade, increasing attention has been given to UFP exposure and its direct effects on the cardiovascular system. Scientific reports on the individual effects of UFPs provide data on blood markers of CVDs, blood pressure, and electrocardiogram (ECG) parameters. Studies were conducted under laboratory, occupational, or real-world conditions. Laboratory and occupational studies generally involved small populations, while various real-world experiments were conducted within the framework of retrospective or prospective cohort studies.

A previous systematic review on the role of UFPs in cardiovascular conditions focused on a quantitative estimation of short-term ambient UFP effects on HRV indices across varying time spans. Their analysis showed an overall trend of reduced HRV parameters following UFP exposure, mainly as a rapid response within hours [7]. It is well known that reduced HRV parameters directly affect BP. Moreover, reduced vasodilation of resistance vessels, due to chronic exposure to air pollutants, partially explains the increase in BP. However, the literature lacks a systematic review on the effects of ambient UFP exposure on BP.

The aim of our systematic review was to provide, for the first time, quantitative estimates of both short- and long-term ambient UFP effects on SBP and DBP. Particular attention was given to studies that used multi-pollutant models to isolate the effects of UFPs from those of other pollutants. Here, we present a synthesis of experimental and observational epidemiological data published in the last 10 years on the effects of UFPs on blood pressure.

## 2. Methods

This review was prepared in accordance with the Preferred Reporting Items for Systematic Reviews and Meta-Analyses (PRISMA) 2020 statement [8].

### 2.1. Eligibility Criteria

The eligibility criteria were established using the PECOS model (Appendix A). The article search was based on the keywords: “nanoparticles” (and its synonym: “ultrafine particles”), “cardiovascular” (the main outcome of interest), and “exposure” (as the primary cause of the outcome). Only experimental research articles were included in this study, while reviews (including systematic reviews), book chapters, meta-analyses, comments, and letters were excluded. Studies from conference proceedings or grey literature were also excluded. The focus of this study was on experimental research with direct participation of humans (e.g., cohort studies, cross-over studies, cross-sectional studies, panel studies, and longitudinal studies) or indirect participation of humans (ecological studies). Animal studies, animal models, as well as in vitro and ex vivo studies using human and/or animal tissues and/or cells were excluded from this investigation.

Articles that met the following criteria were analyzed in our review: (1) epidemiological studies investigating at least one of the following BP indices: SBP and/or DBP; (2) studies evaluating UFP measures for a size range with a lower limit of ≤10 nm; (3) studies analyzing short-term UFP effects with a lag of ≤7 days; (4) studies analyzing long-term UFP effects with a lag of ≥7 days; (5) studies reporting quantitative measures of associations with 95% confidence intervals (CIs) or standard errors from single-pollutant models; (6) studies written in English.

All selected studies focused on a clear relationship between UFPs and BP variability. Both studies with and without statistically significant correlations between UFPs and BP were analyzed in this systematic review. Only studies quantifying UFPs (diameter ≤ 100 nm) were considered. This study focused on the short- and long-term effects of UFP exposure leading to BP imbalance. Thus, only articles presenting high-quality data on BP variability as an effect of UFP exposure were selected for this systematic review.

#### Information Sources

The records were retrieved from Web of Science, Embase, PubMed, and Scopus databases, with eligibility assessed based on titles, abstracts, and full texts when necessary. All databases were last searched on 9 October 2024.

### 2.2. Search Strategy

The search strategy was based on the keywords: “nanoparticles” OR “ultrafine particles” AND “cardiovascular” AND “exposure”.

All selected studies were published in English (language filter) between 1 January 2013 and 9 October 2024 (date range filter).

### 2.3. Selection Process

Two reviewers (JIL and PG) separately screened the records gathered from the databases, assessing each record’s eligibility based on titles, abstracts, and full texts when necessary.

Studies that satisfied the following criteria were included in this review: (1) epidemiological studies investigating BP indices—SBP and DBP; (2) studies reporting UFP measures represented by particle number concentration (PNC) for a size range with a lower limit of ≤10 nm; (3) studies investigating both short- and long-term UFP effects; and (4) studies presenting quantitative measures of associations with 95% CIs or standard errors from single- and multiple-pollutant models.

In the next step, studies that did not analyze quantitative effect estimates standardizable for meta-analysis were excluded.

### 2.4. Data Items and Data Collection Process

Two reviewers (JIL and PG) separately extracted the following data from the records using a Microsoft Excel Open Office 360 template: (1) first author and year of publication; (2) study design, location, study periods, and population; (3) outcome and exposure indices, along with statistics and exposure assignment methods; and (4) effect estimates of UFP with their corresponding 95% CIs.

### 2.5. Study Risk of Bias Assessment (RoB)

The risk of bias (RoB) assessment tool developed by WHO (WHO, Risk of Bias Assessment) was applied in this study. Briefly, RoB was assigned to five areas (Appendix A): confounding, selection bias, exposure assessment, outcome measurement, and missing data. Each area consisted of 1–4 domains. JIL and PG scored the RoB for every domain as “low” risk, “some concerns”, or “high” risk and provided the rationale for all assignments in a Microsoft Excel template.

RoB was scored as “high” if essential data were not provided or if the data were not properly assessed. It was scored as “some concerns” if data were not provided within the article or its references. If any domain received a “high” RoB rating, the entire area was rated as “high” RoB. If any domain had a “some concerns” rating and no domain was rated “high” RoB, the area was rated as “some concerns” RoB. An area was rated as “low” RoB only if all domains had a “low” RoB score.

Any disagreements in record selection, information extraction, or RoB assignment were resolved through discussion between JIL and PG.

### 2.6. Sensitivity Analysis

We conducted the following sensitivity analyses to explore the potential sources of heterogeneity and test the robustness of the pooled effect estimates: (1) We restricted our meta-analyses to BP quantitative measurements with 95% confidence intervals (CIs) effect estimates.

(2) We included only studies that assessed ultrafine particles (UFP) for a size range with a higher limit of ≤100 nm, following the UFP definition suggested by the WHO Good Practice Statement on quantifying ambient UFP (“WHO Global Air Quality Guidelines”, n.d.); (3) We separately pooled short- and long-term effect estimates on systolic blood pressure (SBP) and diastolic blood pressure (DBP). (4) In all analyses, we selected the effect estimate that was the most statistically significant for each study population and indicated a change in blood pressure (BP) in association with elevated particle number concentration (PNC).

To estimate the short-term effects of UFP on SBP and DBP, we conducted additional meta-analyses pooling effect estimates of UFPs on both SBP and DBP.

We used Statistica (StatSoft Polska Sp. z o.o. 2024; Zestaw Plus wersja 5.0.96; www.statsoft.pl, Microsoft Excel OpenOffice software, MetaXL packages, and Grapher 7.01870 Golden software) for statistical analyses and the generation of related plots.

In this systematic review, we focused our analysis on the short- and long-term time effects of BP. Thus, the sensitivity analysis omitted other aspects, which could influence the heterogeneity of the results. Moreover, a small number of the available records were not included in the sensitivity analysis of the dose–response effects.

It is worth noting that the methods of PNC determination vary among studies. Thus, the precision and accuracy of the reported UFP concentrations depend on the quantitative method chosen by the authors. Computational models chosen for cohort studies, covering long periods and wide areas without air-quality monitoring systems with particle counters, are less accurate and less precise than air-monitoring systems chosen for cross-sectional studies. However, air-monitoring systems quantify only outdoor air pollution. Personal monitoring is the most precise and accurate method to quantify both indoor and outdoor PNC, and it was chosen for short-term effect studies with a small number of participants.

The analyzed studies were conducted on different continents and countries with different industrialization levels. Thus, the baseline air pollution was significantly different among different studies. As expected, the dose–response effect was influenced by the basal pollution.

### 2.7. Effect Measures and Statistical Analysis

The effect estimates expressed as mean changes in blood pressure (BP) (95% CIs) or % changes in BP were taken from the original manuscript. If the final results of the effect estimates were referenced in different formats, the proper standardization of the effect estimates was calculated (Appendix A). We were unable to express all research results in mean changes in BP or % changes in BP due to missing data in the original manuscripts.

We conducted meta-analyses on BP indices when at least three effect estimates were available for a specific exposure time (short- or long-term). In each analysis, we selected the lag time showing the most statistically significant effect per study population, regardless of the effect direction.

The Q-test and I² were calculated to measure heterogeneity among studies according to the Neyeloff et al. method [9]. We were unable to execute the funnel plot analysis with Egger’s and/or Begg’s Test due to an insufficient number of results (minimum 10 results for each test).

The mean effect estimates of SBP (mmHg), ∆SBP, DBP (mmHg), and ∆DBP values were referenced from the selected records (from the main text and/or tables presented in the main part of the article or in the electronic Appendix A). Missing data were recovered from the presented figures using the open-source PlotDigitizer software (PlotDigitizer: https://plotdigitizer.com/app; last access on 6 November 2024).

### 2.8. Synthesis Methods

The studies analyzing mean systolic blood pressure (SBP) and mean diastolic blood pressure (DBP) changes in adult participants were selected for the overall SBP and DBP analysis. There was one study [10] analyzing SBP and DBP changes following UFP exposure in children, which was not included in the overall analysis and meta-analysis due to significantly different SBP and DBP baseline values compared to the values in adults. The study by Liu et al. [11] was excluded from the overall analysis because the data provided by the authors were not sufficient to calculate the mean SBP and DBP changes.

## 3. Results

### 3.1. Study Selection

Our literature search in four databases (Embase, Web of Science, PubMed, and Scopus) retrieved 3855 records (Figure 1). After removing 811 duplicates, we analyzed 3044 records and selected 19 eligible articles, which were further used for blood pressure (BP) analysis. Table 1 shows the main characteristics of the records used for the BP systematic review analysis.

### 3.2. Study Characteristics

Studies examined for blood pressure (BP) analysis were conducted in China (4), the U.S. (5), the EU (5), Canada (2), Taiwan (1), Australia (1), and Kazakhstan (1). Sixteen studies analyzed ultrafine particle (UFP) exposure in real-world conditions (indoor and/or outdoor), while three studies [13,18,23] were conducted in laboratory conditions, measuring the particle number concentration (PNC) of UFPs generated during controlled processes (e.g., candle burning, beef cooking).

Among the 19 analyzed studies (Table 1), twelve cross-over, cross-sectional, and panel studies focused on the short-term BP effects of UFP exposure, while seven panel, cross-sectional, cohort, and longitudinal studies presented the BP effects of long-term exposure.

There were only three cross-over studies measuring blood pressure (BP) of each participant before and after exposure at regular time intervals [16,18,23], and these studies were selected for the meta-analysis of BP changes at different time points following short-term ultrafine particle (UFP) exposure.

Among the 19 studies, there was only one study involving healthy children [10]. Two studies focused exclusively on either healthy female [12] or healthy male [19] participants. In one panel study [15], more than half of the 220 participants were overweight; one cross-over study enrolled 34 middle-aged individuals with metabolic syndrome [27], while cohort [20], longitudinal [25], and cross-sectional [21] studies examined entire populations without discriminating based on health status. The remaining studies enrolled only healthy adult volunteers.

In most of the analyzed studies (except for [21,25]), various air pollutants (e.g., PM_2.5_, PM_10_, nitrogen oxides, sulfates, ozone) and environmental parameters (e.g., temperature, humidity) were measured (Table 1). However, there is no consistency in data collection and analysis across these studies. The methods of PNC determination vary among studies influencing their precision and accuracy and increasing heterogenicity among analyzed records.

Only four studies used personal UFP exposure assessment [12], and two studies used a monitoring chamber [13,18]. Two studies utilized air modeling systems to estimate the particle number concentration (PNC) of UFPs [20,25], while the remaining studies used stationary particle counters in the area of interest (Table 1). Moreover, the baseline PNC concentrations vary significantly among studies due to differences in economic and sociocultural levels among studied populations.

#### 3.2.1. Short-Term Effect Studies Characteristics

The statistical data calculations among the selected studies were particularly heterogenous. Liu et al. [11] analyzed short-term health outcomes caused by environmental factors (UFPs, PM_2.5_, SO_2_, NO_2_, O_3_ and CO) in two experimental areas (steel plant as exposed site, and campus area as unexposed site) by applying the non-parametric Wilcoxon Scores Rank Sums test. They used mixed-effects regression models to calculate the differences of health outcomes between two experimental areas, and the relationship between health outcomes and exposure to outdoor air pollutants.

Also, Weichenthal et al. [12] applied linear mixed-effects models with random-subject intercepts to calculate the association between short-term air pollution exposures and health outcomes. Other covariates considered were as follows: age, race, caffeine consumption, alcohol consumption, recent illness, and second-hand smoke exposure. They did not analyze potential effect modification by body mass index (BMI; few participants were classified as overweight or obese).

Devlin et al. [13] used linear mixed effects models to quantify changes in BP of each individual before and after exposures to UFPs. Random intercept models were calculated for the individual baseline values and allowed distinction of within from between-subject variability. A few confounding parameters were considered (e.g., age, gender, medication use, diurnal variation, etc.).

Linear mixed-effects regression models with subject-specific random intercepts were used by Meier et al. [14] to estimate exposure-related health effects clustered in individuals. Separate models were used for each particle exposure metrics and included separate continuous variables for noise. All calculated models were adjusted for age and BMI (continuous variables). Confounding of other air pollutants (temperature, humidity, NO_2_, O_3_, CO) was assessed by sensitivity analyses.

Kubesch et al. [16] applied mixed effect models to calculate the intra-individual variability in short-term exposure on UFPs and other air pollutants (Table 1). They also considered: age, BMI, gender, on-site temperature and relative humidity, and the time of measurement. Moreover, the following covariates were included: exposure time of each participant on environmental tobacco smoke (estimated value), the energy expenditure of each participant during physical activities, and the participants’ exposure on nitric oxides for the day before.

In the study by Padró-Martínez et al. [17] the differences in BP before and after exposure were assessed with a paired *t*-test. Moreover, tests of the sensitivity to PNC exposure at different exposure times were prepared.

Soppa et al. [18] analyzed the short-term exposure on PM_10_, PM_2.5_, PNC, and changes in BP by multiple-mixed linear regression with a random participant intercept. The covariates considered in the full model were as follows: gender, age, weight, height, travel time before exposure, mode of transportation, mean humidity and temperature in the chamber. Finally, they examined how the exposure duration influenced the personal cumulative exposure, thus changes in BP.

Schubauer-Berigan et al. [22] used multiple linear regression to model the correlation between the exposure metrics and BP outcomes, adjusted for confounders (age, sex, race/ethnicity, cigarette pack-years, current or past occupational exposures to chemicals and nanoparticles, childhood pneumonia, current self-reported respiratory diseases, alcohol consumption, use of certain medications). Potential covariates were treated as continuous or categorical variables.

In the study by Gabdrashova et al. [23] three-way ANOVA statistical test was utilized to test relationship between three independent categorical variables (food consumption, exposure chronology and exposure status) with the short-term continuous dependent outcome variable (three separate analyses for the three different dependent outcome variables: heart rate, SBP, and DBP) and any two-way and three-way interplay among the three independent variables for the continuous dependent outcome variable.

In the article by Lyu et al. [25] continuous variables were presented by mean and standard deviation. The unpaired Student’s *t*-test was used to calculate the statistical significance of differences in selected short-term effects parameters between the control group and the exposed group. Analysis of variance was performed to differentiate between multiple groups, and bonferroni method was selected for pairwise comparison of multiple groups. The Chi-squared Test and Fisher’s exact probability methods were applied to test the statistical significance of differences in determined parameters between the control and the exposed group. Multiple linear regression analysis was chosen to analyze the association between exposure status, age, gender, BMI, smoking and drinking habits, and working age in printing rooms.

Guo et al. [26] calculated spearman correlations to evaluate the relationship between studied pollutants before including them in the regression analyses. Linear mixed models were used to calculate the associations between UFPs and BP. The selected covariates included (as fixed-effect terms): age, sex, BMI, exercise times per week, average temperature, and humidity.

In the study by Gilbey et al. [28], the association between UFP and BP variables were determined by Pearson correlation coefficients. Multiple regression analysis was calculated to evaluate the effect of UFP exposure on BP.

#### 3.2.2. Long-Term Effect Studies Characteristics

Chen et al. [19] used linear mixed-effects (LMEs) models to estimate the relationship between acute long-term cardiopulmonary outcomes upon exposure to PM_2.5_ and its constituents. The levels of the biomarkers were considered as the dependent variables, while moving average concentrations of PM_2.5_, black carbon (BC), and PNC of UFP were considered the independent variables. All the LME models were fixed for ambient temperature and relative humidity.

Chung et al. [15] used maximal likelihood mixed-effects, repeated-measures models to associate long-term effects on SBP and/or DBP with daily average of ambient PNC, PM_2.5_, or BC. Mixed models were using a random intercept for each participant and an unstructured (assumption free) covariate matrix structure.

The long-term relationship between the air pollutants and biological markers of cardiopulmonary diseases was studied by Liu et al. [21] by summary statistics for air pollution and health data, Pearson’s correlation for air pollutants and mixed-effects models. The dependent variables were as follows: SBP, DBP, Forced Expiratory Volume in 1 s (FEV1) and High-sensitivity C-reactive protein (hs-CRP). The independent variables were: UFPs, PM_10_, PM_2.5_, nitric oxides and O_3_. The confounding variables were: age, gender, BMI, day of week, season. The daily temperature and humidity were fitted by using a penalized cubic regression spline model.

In the article of Pieters et al. [10] the long-term relationship between air pollution and BP was investigated by calculating the pollutants as continuous variables. The mixed models were adjusted for: age, gender, height and weight of the child, heart rate, type of school, neighborhood type, socioeconomic status, parental education, fish consumption, day of the week, season, wind speed, relative humidity and temperature.

Roswall et al. [27] studied both short- and long-term relationship between air pollution and lipid levels and BP by the use of multivariate linear regression models. Model 1 was fixed for age and gender; while model 2 was fixed for education, marital status, income, smoking status and time since last smoke before blood sampling, environmental tobacco smoke, alcohol before blood sampling, hours of physical activity practicing, BMI, and area-level variables on percentage of the population with low income, percentage with only basic education and percentage in social housing, as well as green space at the residence.

In the article by Corlin et al. [20], two different multilevel linear models for long-term SBP, DBP, pulse pressure (PP) and ln(hsCRP) outcomes were developed to consider the longitudinal relation to PNC. The covariates (gender, medication use, family history of hypertension (for BP), family history of CVD (for hsCRP), diabetes, smoking, employment status at baseline, physical activity, age, and BMI) were selected in a multi-stage process.

Lin et al. [24] divided the participants into three groups, namely hypertension, prehypertension and normotension. Statistical significance of long-term effects was calculated with the use of one-way analysis-of-variance test for continuous variables, and with the use of chi-square tests for categorical variables. Model 1 was fixed for a community-level random intercept. Model 2 was fixed for age, gender, educational level, yearly household income and ethnicity. Finally, model 3 was fixed for lifestyle factors, such as smoking behaviors, regular exercise, controlled diet, BMI and season of BP measurement.

### 3.3. Risk of Bias in Studies

The RoB assessment was prepared according to the WHO guidelines [29], and the results are shown in Figure 2 (results relative to meta-analysis) and Appendix A (detail data on all selected studies). Overall, the selected records were of satisfying quality, and the relationship between UFP and BP was properly assigned. In the ‘confounding’ domain (D1) only one study was rated as “high” risk, and three studies were evaluated as “some concerns” risk. The risk was assigned “high” when there was a lack of analysis of critical potential confounders (or the presented analysis was insufficient); the potential confounder was not validly assessed; or when authors used unsuitable methods or experimental designs when fixing for critical and other/additional confounders. In the ‘selection bias’ domain (D2), four studies had “some concerns” risk assignment (when participants did not have the same opportunity to be in the study, but did not bias the effect estimates). In the ‘exposure assessment’ domain (D3), 6 out of 19 studies had “some concerns” risk assignment, while the remaining 13 studies were evaluated as “low” risk. The risk was assessed “some concerns”, when exposure levels were assessed less properly (but did not bias the effect estimates), measurement methods varied across the range of exposure (but there was proof that the exposure measurement is enough similar and that effect estimates were not strongly biased), and/or spatial exposure contrasts could have changed through the experiments and were not taken into account (but effect estimates were not strongly biased), and/or exposure contrast was low relative to the within-subject variance (but not to the extent that the study is uninformative). There were three studies with “some concerns” risk in the missing data domain (D5), while the remaining studies were evaluated as “low” risk. The “some concerns” risk was assessed when missing data on outcomes had low frequency (≥10%). All 19 studies were evaluated as “low” risk in outcome measurement domain (D4).

### 3.4. Results of Individual Studies

#### 3.4.1. Short-Term Effects Individual Studies

The literature search in the last 11 years (2013–2024) showed twelve studies on short-term effects of UFPs exposure, among which only two experimental studies evaluating short-term BP changes under controlled increase of UFPs [18,23].

In the study by Soppa et al. [18], 54 healthy volunteers (men (27) and women (27); aged 18–79 years) were exposed for 2 h to UFPs (mean PNC ranging 600–2670 × 10^4^ 1/m^3^) generated in three different scenarios: candle burning, toasting bread and frying sausages. PNC growth of 50,000 particles/cm^3^ was related to the increase of SBP at all time points, except for the point post-exposure. Directly after UFP exposure, SBP was increased by 0.3 mmHg SBP (95% CI: −0.1; 0.7), while 4 h after exposure, SBP was increased by 0.8 mmHg SBP (95% CI: 0.5; 1.2). The growth of UFP exposure metrics during candle burning and frying sausages was not related to changes in BP.

The study by Gabdrashova et al. [23] enrolled 17 healthy non-smoking adults (7 male and 10 female participants, aged 18–46 years) for the phase 1 study, while in the phase 2 there were 33 non-smoking healthy participants (11 men and 21 women, aged 18–51 years). In the phase 2, there were 16 participants that could eat and drink during the study, while the remaining participants were not allowed. The controlled exposure on beef frying-emitted UFPs lasted for 20 min. According to the authors, there were no statistically significant DBP and SBP differences (a two-sided α > 0.05) in the Phase 1 study, when compared to the before-cooking timespan. Statistically significant SBP growth was found during the phase 2 study. The authors noted that the fluctuations or SBP lowering were related to the absence of food/drink. When the entire phase 2 study population was analyzed, there were no significant changes in SBP.

Devil et al. [13] focused on the U.S. population with metabolic syndrome with and without null GSTM1 (a prominent antioxidant gene) allele. In their randomized cross-over study, 34 participants were exposed for 2 h to clean air or concentrated ambient UFPs in laboratory chamber. The authors reported no change in blood pressure immediately following UFPs exposure or the next morning.

Meier et al. [14] presented the panel study with the enrollment of 18 healthy highway maintenance workers exposed to air pollutants for five consecutive days. Air pollutants and noise after work were positively associated with SBP and DBP pressure the next morning, whereas work noise was not associated significantly with blood pressure.

In the cross-sectional study conducted by Lyu et al. [25], there were 107 participants (aged > 16 years) enrolled, among which 53 were printing room workers (exposed group), while the remaining participants were considered the control group (unexposed group). The PNC were quantified using a stationery UFPs counter, which was placed for more than 5 min in the print shops and workplaces of the control group. The presented results showed that SBP (−3.36; *p* = 0.001) and DBP (−3.02; *p* = 0.002) values increased significantly in the group of exposed workers compared to the group of unexposed controls (unpaired student-*t*-test: *p* < 0.05).

Schubauer-Berigan et al. [22] studied occupational exposure of 108 workers on carbon nanotubes and nanofibers. Even if carbon nanotubes concentration in the air was low, it was observed that SBP was positively associated with UFPs (*p*-values: 0.015–0.054).

The panel study of Guo et al. [26] was conducted with the participation of 88 healthy university students in Guangzhou (China). Five weekly measurements of BP showed that PM of all the fractions in the 0.2 to 2.5-μm range were positively correlated with SBP after one day (24 h), with percentage changes of effect estimates ranging from 3.5% to 8.8% for an interquartile range increment of PM. Moreover, PM0.2 was positively correlated with DBP in the time range from 7 to 12 h but were not significant above 1 day (24 h).

Gilbey et al. [28] studied the association of UFPs exposure and BP changes in the indoor environment. The results obtained with the participation of 40 healthy volunteers showed that PM_2.5_ was associated with an increase in DBP (3.2 mmHg; 95% CI: 0.99, 5.45) rather than UFP fraction.

Crossover real-world exposure study with participation of 28 healthy participants was made by Kubesch et al. [16] in Barcellona. In the study, SBP and DBP responses of each participant after four different exposure scenarios were measured. The study showed that exposure to high- traffic-related air pollution (TRAP) was related to post-exposure DBP increase (1.1 mm/Hg, *p* = 0.002), irrespective of physical activity. Interquartile range (IQR) growth of UFPs was related to statistically significantly post-exposure SBP increase (1.1 mm/Hg). Intermittent physical activity compared with the rest time span was related to lower SBP post-exposure (−2.4 mm/Hg, *p* < 0.001). Physical activity decreased SBP more after exposure to the low-TRAP area (−2.3 mm/Hg), rather than after exposure in high-TRAP area (−1.6 mm/Hg).

The randomized cross-over study, with the participation of 61 healthy, non-smoking volunteers (54% females, median age 22 years) was executed by Liu et al. [11]. Each participant spent randomly 5 consecutive days (8 h outdoor) in a residential area close to a steel plant (exposure area), or on a college campus (5 km away from the plant). The washed-out period lasted 9 days. During mid-day, each participant practiced 30-min moderate physical activity. BP was measured daily and post-exercise at both study areas. The stationery particle counter was used for PNC quantification. The obtained results showed that SBP and DBP measured during the rest time, and post-exercise were not significantly different between two areas.

Padró-Martínez et al. [17] conducted a cross-over trial of high-efficiency particulate arresting (HEPA) filtration in public housing near a highway. The results obtained with the participation of 20 healthy residents, living in the close proximity to the highway, showed that UFPs exposure was correlated with BP decrease immediately after exposure (SBP mean change and 95% CI: β = 0.32; −4.93–5.57; DBP mean change and 95% CI: β = 2.73; −2.66–8.12), after 24-h (SBP mean change and 95% CI: β = −5.92; −12.3–0.51; DBP mean change and 95% CI: β = −4.45; −8.45–(−0.45)), and 48-h (SBP mean change and 95% CI: β = −4.92; −15.2–5.29; DBP mean change and 95% CI: β = 1.17; −7.16–9.50).

The Weichenthal’s et al. [12] enrolled for the cross-over study only female participants (53 volunteers). Women were exposed to traffic pollutants for 2-h in three different scenarios, namely cycling on high-traffic routes, cycling on and low-traffic routes, and cycling indoor. Personal UFPs exposures on traffic pollutants were correlated with BP changes immediately following exposure to high and low-traffic. The results showed the positive correlation between UFPs exposure and DBP (1.61; 95% CI: −0.155, 3.38).

#### 3.4.2. Long-Term Effects Individual Studies

According to our literature research there were seven studies in the last 11 years focusing on the long-term effects of UFPs exposure, among which four panel studies, one longitudinal study [20], one cross-sectional study [24] and one cohort study [27] analyzing the overall effect of UFPs generated in real-environment scenario on the entire populations.

In the study by Roswall et al. [27], long-term (7, 30 and 90 days) effects on UPFs exposure were evaluated on 32,851 Danes (aged > 18 years) from the Diet, Cancer and Health—Next Generations cohort. The exposure on air pollutants (among which UFPs) was modelled by means of the Danish DEHM/UBM/AirGIS modelling system. For UFPs, the authors found higher SBP values 2.45 (β-estimates = 1.70; 3.12) and DBP values 1.56 (β-estimates = 1.07; 20.4) per 10,000 particles/cm3. Of note, the β-estimate showed to get higher when time of exposure was longer.

Corlin et al. [20] analyzed an UFPs effect on Boston Puerto Rican Health Study population. Residential annual average UFP exposure was calculated with the use of spatial and temporal trends model. The 791 adult participants (69% female) were visited up to three times in six years (2.2 years’ time between 1st and 2nd visit). The authors calculated that the overall, long-term PNC exposure was not related to SBP (β = 0.96; 95% CI = −0.33, 2.25 mmHg per 4600 particles/mL). However, when the SBP baseline level was considered, PNC was related to the changes in SBP (β = 1.66; 95% CI = 0.17, 3.14 mmHg). Moreover, PNC was positively correlated with SBP in never smokers’ group (β = 2.20; 95% CI = 0.04, 4.37 mmHg), but not in the group of former or current smokers. There was no overall association between PNC and DBP (β = 0.55; 95% CI = −0.20, 1.30 mmHg per 4600 particles/mL), with the exception of never smokers’ group (β = 1.32; 95% CI = 0.19, 2.46 mmHg).

Lin et al. [24] enrolled 24,845 adults living in Northeast China for the cross-sectional study. The average concentrations of UFPs were calculated with the use of chemical transport model. According to the authors, one unit (1 μg m^−3^) growth of UFPs was correlated with SBP growth of 1.52 mm Hg [95% CI: 0.48–2.55], and DBP growth of 0.55 mm Hg (95% CI: 0.01–1.08). The relationship between UFPs and SBP (and DBP) were higher in women and non-drinkers group.

In the panel study made by Liu et al. [21], 100 norther Taiwan inhabitants (50% female) were enrolled for the study. The stationary particle counter was used to quantify for 24 h (before 8 a.m. and 8 a.m. the day after) the UFPs concentration in each participant’s apartment before measuring the participants’ BP. The authors reported that an IQR growth in the 24-h mean UFPs (0.97 μg m^−3^) was related to a 6.3% SBP growth [95% confidence interval (CI) = 2.9, 9.7] and 5.6% DBP growth (95% CI = 4.1, 7.1).

Pieters et al. [10] conducted their studies on 130 healthy children aged 6–12 years old, measuring blood pressure of each participant during two seasons (spring and fall). The authors reported that for the total UFP fraction, SBP was 0.79 mmHg (95% CI: 0.07, 1.51; *p* = 0.03) higher after exposure. Any SBP effects were observed after the exposure on UFP fractions. Of note, the increase of UFP fraction (20–30 nm) was related to a 6.35 mmHg (95% CI: 1.56, 11.14; *p* = 0.01) SBP growth. DBP was not correlated with any of the studied particulate mass fractions.

In the study of Chung et al. [15], 220 participants (mean age 58.5 years) from the Community Assessment of Freeway Exposure and Health study (most of the participants were living close to main highways) were enrolled for the panel study. Ambient PNC concentration was measured by the stationary particle counter placed on the roof of the building located near the volunteers’ residence. The presented results showed that PNC higher than 10,000 particles/cm3 was related to DBP increase of 2.40 mmHg (*p* = 0.03), particularly in obese individuals.

Chen et al. [19] recruited in Beijing (China) 20 healthy non-smoking male volunteers for the panel study. The baseline questionnaire was used to collect data on exercise preferences and frequency. The outdoor PNC were registered with the use of fixed monitoring station. The results of a stratified analysis showed that acute cardiopulmonary responses were influenced by physical activity habits. An IQR growth of 9–13-d UFP moving average exposure was correlated with a 13–17% growth of aortic augmentation pressure in participants who chose outdoor exercise. SBP growth was correlated with 5- and 7-d UFP moving average exposure to accumulation mode particles (β = 1–2%).

### 3.5. Meta-Analysis

In the first step, the overall analysis (both short- and long- term effects upon UFPs exposure) of mean BP values was prepared. Considering huge variability of experimental design and statistical analysis among 19 studies taken in the presented analysis, we selected 17 records for the overall mean BP values analysis (Figure 2). Two studies were excluded from the results of synthesis analysis due to low age of the participants [10] or inability of mean BP data extraction [11]. Figure 2 shows the box plot of mean SBP and DBP values collected in 17 studies evaluating short- and long-term effects of UFPs exposure. We can observe that IQR of mean SBP values is wider than the IQR values of DBP. The SBP values of short-term exposure are lower than long-term exposure values (short-term exposure: mean 115.7 mmHg, median 118.1, IQR 17.2; long-term exposure: mean 122.7, median 123.9, IQR 18.5). On the contrary, the short-term exposure DBP values are similar to long-term exposure DBP values (short-term exposure: mean 73.7 mmHg, median 75.0 mmHg, IQR 7.7 mmHg; long-term exposure: mean 74.1 mmHg, median 73.7 mmHg, IQR 5.4 mmHg). Of note, the long-term mean SBP value exceeds the SBP normotension limit (≤120 mmHg) [30].

It is interesting to observe that the ratio of the mean SBP to the mean DBP values is lower and narrower after short-term exposure than after long-term exposure (Appendix A).

In the next step, the short-term and long-term effects of UFP exposure on SBP (Figure 3A,C,E) and DBP (Figure 3B,D,F) were analyzed separately. The variability among different experimental approaches and, consequently, statistical analysis of the obtained data let us compare the effect estimates of only eight studies focusing on short-term effects and three studies analyzing long-term effects. The main characteristics of the studies included in the systematic review but not in the meta-analysis, with the reason for exclusion, are presented in Appendix A.

We can observe (Figure 3) that according to the analyzed data, the UFPs’ exposure is positively correlated with BP both in the short- and long-term. Figure 3A,C present the effect estimates of short-term SBP changes after exposure, while Figure 3B,D show the effect estimates of short-term DBP changes. The long-term effects on SBP and DBP changes are presented in Figure 3E and Figure 3F, respectively.

There is significant heterogeneity (expressed as I^2^) among analyzed short-term exposure data due to the use of different experimental protocols (Table 1) in selected studies. The selection of studied populations (different age and gender distribution), accuracy in UFP quantification (personal exposure assessments, fixed monitoring stations, air quality models, laboratory conditions or real-world environments), UFP concentration, time of BP measure after exposure, and the choice of confounding factors (among which other air pollutants) used in the analysis of effect estimate are influencing mainly the heterogeneity. Moreover, there are various types of study designs, such as cross-sectional studies, cohort studies, panel studies, and cross-over designs. Some studies focus on short-term effects, while others focus on long-term effects (seven studies). Of note, study durations range from hours to years.

Due to high heterogeneity, the effect summary models for short- and long-term effects on SBP and DBP were calculated with both fixed-effects and random-effects models (Figure 3). In addition, the graphical representation of RoB analysis is provided in Figure 3.

Among studies analyzing short-term effects of UFPs’ exposure (Figure 3A–D), most of the presented results have narrow CI, except for Gilbey et al. [28], Padró-Martínez et al. [17] and Guo et al. [26] Comparing the experimental protocols (Table 1), it can be observed that studies with the narrow CI collected the BP results punctually and in the short time after exposure (within few hours). On the contrary, the BP data of the studies with large CI were collected less precisely, with the collection range between one day and a few days. It is likely that outcome results of the selected studies were highly influenced by the data collection time after exposure.

Table 1 shows that mean PNC values vary significantly among selected studies. Higher outdoor [12,13,14,16,20,25] PNC were registered in highly populated and industrialized areas. Moreover, there is also a significant variability of PNC quantitative methods, with different precision and accuracy, used in selected studies. The most precise and accurate is personal monitoring method measuring both indoor and outdoor UFPs concentrations. However, personal monitoring strategy is limited to the low number of participants and in a short-time period. The significant variability of PNC data enabled the reliable dose-repose analysis.

The meta-analysis of the long-term effects of UFPs’ exposure on BP changes was prepared with data from only three studies (Figure 3E,F). The calculated heterogeneity (I^2^) of the selected studies is low. Among selected studies of long-term effects, there is the investigation of Roswall et al. [27] with the data of 32,851 cohort participants. Thus, the effect estimates of this study had a CI range narrower than the remaining studies.

The effect summaries calculated with both fixed- and random-effects models show that UFPs’ exposure is positively correlated in the short- and long-term with SBP and DBP. The effect estimates of SBP on long-term exposure are higher than short-term exposure.

In order to analyze the immediate BP changes after UFPs exposure within few hours, the mean values of ∆mmHg SBP and ∆mmHg DBP at following time spans: 0, 0.1, 0.5, 2, 3, 4, 5 and 24 h after exposure were used for the analysis. The 33 data points for ∆ mmHg SBP and 33 data points for ∆mmHg DBP were extrapolated from three independent studies [16,18,23]. Figure 4A shows the box plot of ∆mmHg of SBP and ∆mmHg DBP at increasing time spans after UFPs exposure, while Figure 4B presents the ratio of ∆mmHg of SBP and DBP at each time point after exposure.

The highest changes in BP are observed during exposure (Figure 4A), while the lowest BP changes are registered after 24 h. The longer the time after exposure, the lower the BP changes compared to the baseline BP. The changes in SBP are significantly higher than the changes in DBP. We can observe that after exposure to UFP, DBP values return to baseline values faster than SBP values. Indeed, the ∆mmHg SBP to ∆mmHg DBP ratio (Figure 4B) is the highest 3 h after exposure.

The observed BP changes were not correlated with the PNC of UPFs. However, the studies reported the PNC only during exposure (time after exposure equal to 0), and there was no data available on PNC at each time point after exposure.

## 4. Discussion

Blood pressure (BP) is a vital parameter, relatively easy, painless, and economical to measure autonomously or by qualified health service operators. However, both systolic blood pressure (SBP) and diastolic blood pressure (DBP) values can be influenced by the local environment (e.g., temperature, atmospheric pressure) and simple activities (e.g., drinking, eating, walking). Nevertheless, a large amount of data has allowed us to establish the correct physiological BP value ranges.

Today, guidelines recommend SBP target values below 140 mmHg and DBP values less than 90 mmHg [31,32,33,34] Following Chobanian et al. [35], subjects can be classified based on BP measurements as having hypertension (SBP ≥ 140 mm Hg or DBP ≥ 90 mm Hg), prehypertension (SBP ranging from 120–139 mm Hg or DBP from 80–89 mm Hg), or normotension (SBP of ≤120 mm Hg and DBP of ≤80 mm Hg) [32,36]. However, the latest American Heart Association (AHA) and European Society of Cardiology (ESC) guidelines suggest even lower targets [36,37,38,39].

The influence of chronic BP increases on cardiovascular disease (CVD) development has been the subject of numerous studies worldwide. The offspring epidemiological study, Framingham Heart Study, aimed at better understanding coronary heart disease in the USA, established that BP is directly related to CVD risk [39]. Additionally, it was shown that isolated SBP hypertension is a significant predictor of CVD [40]. Moreover, the Framingham study demonstrated that SBP and DBP have a continuous, independent, graded, and positive association with CVD [41]. Of note, even high-normal BP values (SBP 130–139 mmHg and DBP 85–89 mmHg, or both) were correlated with an increasing risk of CVD [42].

Over 70 years after the Framingham study began, hypertension is still the most significant risk factor for CVD (Global, regional, and national comparative risk assessment, 2018). According to data presented in 2020, age-standardized mortality rates associated with elevated SBP were the highest in Central and Southeast Asia, Eastern and Central Europe, and parts of Africa and the Middle East [43]. For instance, in 2017, high SBP accounted for 2.54 million deaths in China, among which 95.7% were related to CVD [44].

Evidence from epidemiological studies on UFP and cause-specific mortality is still limited. In a recent study by Lanzinger et al. conducted in five central European cities, city-specific effect estimates for Augsburg showed five-day delayed effects of UFP on cardiovascular mortality (6.0% [1.0%; 11.4%]). The authors assumed that non-significant results for the other cities might be due, at least in part, to missing data and insufficient statistical power [45].

Numerous epidemiological and clinical data have shown that the risk of CVD rises even at BP values lower than the clinical threshold for hypertension. In the recent Kailuan study (a cohort of occupational Chinese adults), every 10 mmHg/year growth of cumulative SBP over six years correlated with a 1.3% increase in all-cause mortality. In particular, there was a 1.8% increase in cardiovascular and cerebrovascular outcomes [46]. In the Coronary Artery Risk Development in Young Adults Study, an increase of one standard deviation in cumulative SBP was related to a 73% increase in cardiovascular risk [41].

Elevated BP has been hypothesized as a pathophysiological mechanism linking exposure to ambient air pollutants with the growing risk of cardiovascular morbidity and mortality. Specifically, a review by Langrish et al. [47] on the cardiovascular effects of exposure to particulate matter in the air indicated that short-term exposures were related to significant increases in BP, arterial stiffness, myocardial ischemia, and induced cardiac arrhythmias. Arterial stiffness, a well-known predictor of hypertension, contributes to inward remodeling of small arteries, increasing arterial resistance and BP. Stiffness affects BP by increasing left ventricular systolic load, while excessive pressure pulsatility lowers diastolic pressure [48].

According to Vidale & Campana [43], three pathways lead to CVD development due to UFP exposure. In the classic pathway, inhaled UFPs cause pulmonary inflammation and oxidative stress, leading to plaque vulnerability and rupture (acute exposure), atherosclerosis and systemic inflammation (subacute exposure), and atherosclerosis with metabolic syndrome (chronic exposure). In the alternative pathway, UFPs may directly affect the cardiovascular system by interacting with the vascular endothelium, causing thrombosis after acute exposure, enhanced coagulation after subacute exposure, and atherosclerosis after chronic exposure. In the central pathway, particulate matter and reactive oxygen species activate alveolar receptors linked to nerve endings, altering heart rhythm and variability [49,50]. The central pathway leads to atherosclerosis, an increase in BP (after acute exposure), and HRV changes (after subacute exposure). Epidemiological studies have shown higher hazard ratios for cardiovascular disease during chronic, compared to acute exposure.

The studies presented here showed different BP effects from short-term and long-term UFP exposure. Mean SBP values for populations exposed to UFPs long-term were higher than those exposed short-term. It is important to understand the extent to which repeated and/or prolonged short-term exposure could lead to a permanent BP increase, but literature lacks sufficient studies on this.

The time-response, dose-response and time-dose-response axes are important in understanding the UFPs exposure effects on BP unbalance. However, the current knowledge is limited to PM studies, while studies on both UFPs’ time and dose effects are missing. For this reason, the discussed here results focus on the BP unbalance after UFPs exposure, rather than time-response or dose-response effects. Moreover, each analyzed study used different strategies of PNC determination and different time ranges after exposure. In addition, most studies focused on outdoor pollutants, omitting the impact of indoor pollutants. Thus, we were not able to conduct good quality analysis of dose-response effects. Future studies could reduce observed here heterogeneity by standardizing study designs and exposure assessment methods with particular attention to UFPs dose and time after exposure.

Air pollutants have been shown to cause acute vasoconstriction and rapid changes in the sympathovagal balance, often seen as transient BP changes [51], particularly in SBP. This mechanism can be activated even after brief exposure. Experimental trials showed BP increases minutes to hours after exposure to air pollutants, which could lead to chronic hypertension development [52,53].

The relationship between SBP and cardiovascular risk varies depending on the outcome, such as stroke or myocardial infarction [54], In patients with high cardiovascular risk and those with stable coronary artery disease, reductions in SBP below 120 mmHg and DBP below 70 mmHg were associated with a higher risk of coronary and cardiovascular death.

Our systematic review data (Figure 2) showed that the range of mean SBP values collected from both short- and long-term exposure studies is wide. Most mean SBP values are below the stage 1 hypertension threshold. However, the third IQR of mean short- and long-term SBP exceeds the 120 mmHg threshold. Noteworthily, the range of mean DBP values (Figure 2) is narrow and does not exceed the 80 mmHg stage 1 hypertension threshold.

The data presented here show that both short- and long-term UFP exposure leads to changes in SBP and DBP values, predominantly increasing both. The estimated short-term SBP and DBP effects are in a similar range to the long-term effects. Of note, long-term chronic UFP exposure studies do not analyze the short-term effects adjacent to BP measurement. Another important issue is that classical short-term (<7 days) and long-term (>7 days) time classification in the studies on air pollutant effects on human health may be too simplistic to capture the differences in health effects across different time scales. The future studies should consider more representative timespans.

In the presented data, we focus for the first time on the analysis of short-term SBP/DBP effects after UFP exposure. It can be observed that short-term UFP exposure led to immediate BP changes, starting in the first minutes of exposure. In healthy subjects, SBP and DBP values tend to return to baseline within 24 h. However, SBP decreases more slowly than DBP after exposure. Thus, the SBP/DBP ratio remains altered, particularly a few hours after UFP exposure. Noteworthily, most studies on short-term effects measure BP values after 24 h, without precise time ranges after exposure (Table 1), leading to significant variability in the obtained results. Moreover, it is likely that the impact on BP is much lower 24 h after UFP exposure than immediately following exposure.

The Golden Ratio (GR), which equals 1.618, is defined as the proportion of two sequential Fibonacci numbers. Various cardiovascular measures, such as systolic to diastolic time intervals on electrocardiogram recordings, pulmonary and systemic hemodynamic measures, and end-diastolic to end-systolic diameters of the left ventricle on echocardiography, have gained interest concerning GR or Golden proportions [52,55,56]. Recently, the SBP to DBP ratio (as a reflection of GR) has been evaluated in patients undergoing ambulatory BP monitoring (ABPM) to assess the presence of hypertension [53].

Between October 2020 and March 2021, Atmaca et al. [53] enrolled 254 patients (212 non-diabetic and 42 diabetic) who underwent ABPM. The SBP/DBP ratios were significantly higher in the diabetic group compared to the non-diabetic group across all analyzed time spans (1.698 vs. 1.631, *p* = 0.041 for 24 h; 1.689 vs. 1.618, *p* = 0.032 for daytime; 1.74 vs. 1.66, *p* = 0.037 for nighttime). Notably, in the non-diabetic group, the daytime SBP/DBP ratio closely resembled the GR value (i.e., 1.618). Moreover, the 24-h SBP/DBP ratio was 1.631. Conversely, in the diabetic group, SBP/DBP ratios deviated significantly from the GR value of 1.618 across all time spans.

The studies by Böhm et al. [50] showed that the mean achieved SBP in diabetic patients (137.9 ± 13.6 mmHg) was higher than in non-diabetic patients (133.5 ± 13.8 mmHg, *p* < 0.0001), while the mean DBP in both groups (78 mmHg) was comparable.

In the present study, the short-term SBP/DBP ratios (Appendix A) were lower than after long-term UFP exposure. The short-term SBP/DBP mean ratio (1.571) was below the GR value, while the long-term SBP/DBP mean ratio was close to the GR value of 1.68.

The studies by Lee et al. [57] highlighted the advantage of joint analysis of SBP and DBP hypertension in predicting CVD death. Their community-based cohort studies on 14,375 patients (categorized into the following subgroups: SBP/DBP hypertension, SBP hypertension only, DBP hypertension only, and non-hypertension) showed that the risk of CVD was most significant in two subgroups: SBP hypertension (HR = 1.59, 95% CI 1.26–2.00) and SBP/DBP hypertension (HR = 1.84, 95% CI 1.51–2.25). The most significant risks of ischemic and hemorrhagic stroke were found in the DBP hypertension group (HR = 4.11, 95% CI 1.40–12.06) and the SBP/DBP hypertension group (HR = 2.59, 95% CI 1.92–3.50). The risk of CVD death was significantly higher in the group with SBP ≥120 mmHg and DBP ≥ 80 mmHg. The most significant CVD risk was observed in groups with SBP of 130–131 mmHg and 134–137 mmHg.

The pediatric cardiopulmonary system is still developing. Therefore, children are particularly vulnerable to inflammation induced by air pollutants. Data collected by the World Health Organization (WHO) suggests that early-life exposure to air pollution may increase the prevalence of CVD in adulthood. Nevertheless, there is a lack of epidemiological studies focusing on cardiovascular effects in children exposed to UFPs in both the short-term and long-term.

## 5. Conclusions

Ultrafine particles (UFPs) are air pollutants that require special attention when assessing the risk of cardiovascular diseases (CVDs) in populations living in areas of high pollution. Chronic UFP exposure could lead to a persistent increase in SBP, even without a corresponding increase in diastolic blood pressure (DBP), leading to perturbations in the Golden Ratio (GR) of blood pressure values, as already observed in diabetic patients.

Under short-term exposure, BP values change according to the exposure duration. The most significant BP imbalance can be observed during and immediately after exposure. Therefore, it is essential to measure BP at different time intervals in experimental studies.

The presented data show how short-term UFP exposure can lead to a significant increase in SBP with a relatively small increase in DBP. Both acute and chronic perturbations in the SBP/DBP ratio may be predictors of cardiovascular risk and require more attention in future research on the short-term and long-term effects of UFP exposure. Considering the simplicity and low cost of BP measurements, accurate and repeated data on SBP, DBP, and their ratio could be part of regular medical visits, together with inquiries about air quality data. However, more studies dedicated to UFPs’ impact on BP and standardization of time–response and dose–response protocols are needed. Particular attention should be paid to vulnerable populations, including children, patients with CVD or diabetes living in air-polluted areas, and patients with previously diagnosed high SBP/DBP ratios.

## 6. Limitations

The main limitation of the analysis presented here is the significant heterogeneity among the 13 studies in terms of experimental protocols and statistical analysis methods. Moreover, only one experimental protocol [17] used both left- and right-arm repeated measurements, highlighting more significant differences in statistical analysis when the mean value of bilateral measurements was used in the model.

There are few experimental studies on the short-term effects of ultrafine particles (UFPs) on blood pressure (BP), and only one [27] considered the effect of eating and drinking during and after exposure. Additionally, there should be more accurate assessments of UFP exposure in everyday life, moving beyond controlled experiments.

In the long-term studies, there is a significant lack of accurate UFP exposure assignment for the participants. Furthermore, there is a critical shortage of air quality monitoring networks that provide high-quality data on major air pollutants, including UFPs. Globally, more attention should be paid to air quality monitoring systems; otherwise, the proper assignment of cardiovascular risks will remain incomplete.

Highly precise and accurate data on air quality, together with PNC quantification, are necessary for future dose–response and time–dose–response analysis. Particular attention should be given to indoor UFP concentrations, while humans are spending most of their time in closed ambient with poor ventilation.

The systematic review presented here is the first attempt to synthesize the available data on blood pressure imbalance after UFP exposure. The available data are not exhausting, and the results obtained are heterogeneous. Thus, further research should focus on individual and combined effects of UFPS and air pollutants on cardiovascular pathologies with in-depth analysis of dose–response and time–dose–response effects. Particular attention should be paid to vulnerable populations.

## Figures and Tables

**Figure 1 jcm-13-06802-f001:**
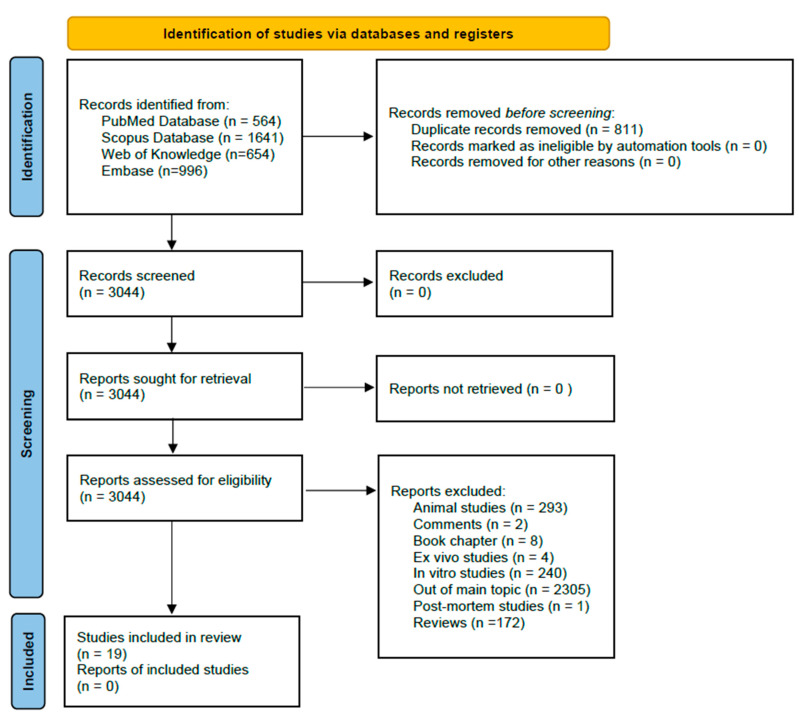
Flowchart of the records screening process.

**Figure 2 jcm-13-06802-f002:**
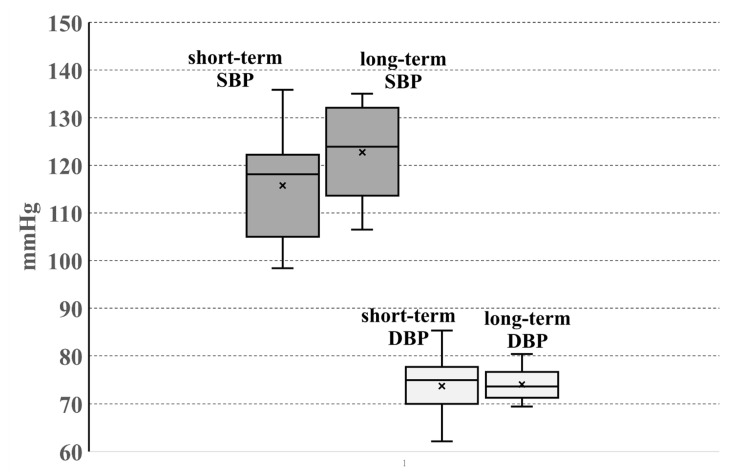
Box plot of mean SBP and DBP values collected in 17 studies evaluating short-term and long-term effects of UFP exposure. × = mean value.

**Figure 3 jcm-13-06802-f003:**
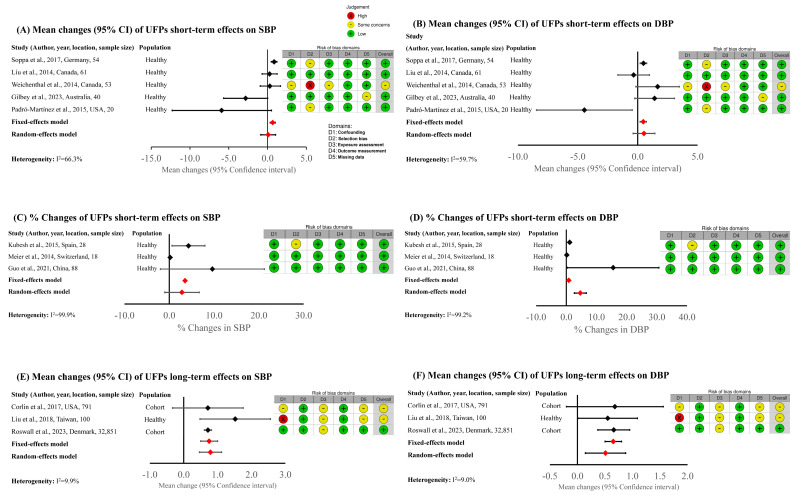
Short-term (**A**–**D**) and long-term (**E**,**F**) effect estimates on mean changes on systolic blood pressure (**A**), mean changes on diastolic blood pressure (**B**), % changes on systolic blood pressure (**C**), % changes on distolic blood pressure (**D**), mean changed on systolic blood pressure (**E**), and distolic blood pressure (**F**) after UFP exposure.

**Figure 4 jcm-13-06802-f004:**
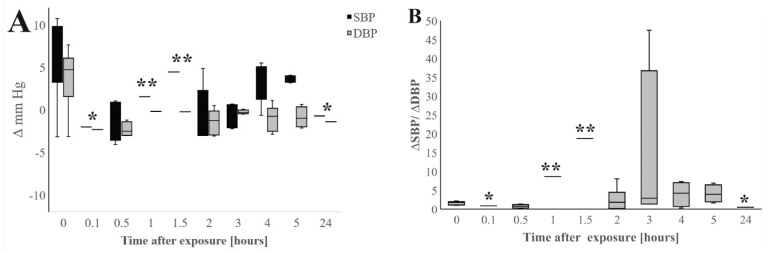
Mean effect estimates for changes (∆mmHg) in systolic (SBP) and diastolic blood pressure (DBP) depending on time after exposure in different exposure scenarios. (**A**) The box plot of mean SBP and DBP changes after 0, 0.1, 0.5, 2, 3, 4, 5, and 24 h after exposure. (**B**) The ∆mmHg SBD to ∆mmHg DBP ratio after 0, 0.1, 0.5, 2, 3, 4, 5, and 24 h after exposure. A statistical comparison of changes (∆mmHg) in systolic (SBP) and diastolic blood pressure (DBP) between different time points was performed using the GraphPad Prism software (version 10.1.0 for macOS, GraphPad Software, Boston, MA, USA). The data were analyzed using the one-way ANOVA test. *p*-value < 0.05 was considered significant. * Data extrapolated from [18]; study conducted on N = 54 healthy participants. ** Data extrapolated from [23]; study conducted on N = 33 healthy participants. * and ** Data were not taken for the one-way ANOVA test.

**Table 1 jcm-13-06802-t001:** Main characteristics of studies included in the systematic review. NA = not applicable.

No.	Study	Study Design	Location	Study Period	Population/Sample Size	Exposure Type	Experiment Type	Exposure Assessment	Size Range (nm)	Mean ± SD (Range) (×10^3^ Particles/cm^3^)	Other Pollutants/Risk Factors	Exposure Measure	Outcome	Time After Exposure of Outcome Measure
1	Liu et al. [11]	Randomized cross-over study	Canada, Sault Ste. Marie	May 2010–August 2010	61 healthy, non-smoking subjects (54% female, median age 22 years)	Outdoor	Real environment	Air quality monitor by fixed-site ambient	10–100	collage site: 4.440 (2.034–15.842); Bayview site: 7.174 (1.415–39.107)	SO_2_, NO_2_ and CO, PM_2.5_, O_3_, temperature, humidity	Continous	Short-term outcome: SBP, pulse rate, DBP, and flow-mediated vasodilation (FMD).	BP and pulse rate were measured daily and post-exercise. FMD was measured at the site near the plant.
2	Weichenthal et al. [12]	Cross-over study	Canada, Montreal	Summer, 2013	53 healthy non-smoking women (mean age 25)	Outdoor	Real environment	Personal	10–100	16.771 (10.302)	PM_2.5_, BC, NO_2_, O_3_, temperature, humidity	2 h	Short-term outcome: reactive hyperemia index, SBP, DBP; HRV.	During cycling (2 h, 11 a.m.–1 p.m.).
3	Devlin et al. [13]	Cross-over randomized	Campus of the University of North Carolina, US	NA	34 middle-aged individuals with metabolic syndrome (13 male and 21 female) mean age of 47.8	Indoor	Laboratory exposure chamber	Monitored in real time	20–250	100	NA	2 h	Short-term outcome: SBP, DBP, brachial artery diameter (BAD), endothelium-dependent flow-mediated dilatation (FMD), nitric oxide-mediated dilatation (NMD), images of the right brachial artery (BAD1); Holter ECG (SDNN, PNN50; HF, LF, premature atrial contractions (PAC), and premature ventricular contractions (PVC)).	Participants were exposed twice, while at rest for a 2 h period: once to clean air and once to concentrated ambient ultrafine particles.
4	Meier et al. [14]	Panel study	Western Switzerland	May 2010 and February 2012	18 participants (healthy highway maintenance workers aged 31–59 years)	Indoor and outdoor	Real environment	Personal monitoring	<100 nm	75.699 ± 81.761	PM_2.5_, noise, and gaseous co-pollutants (CO, NO_2_, O_3_), temperature, and humidity.	continuous	Short-term outcomes: BP, fractional exhaled nitric oxide (FeNO), and lung function, interleukin 6 (IL-6) and tumor necrosis factor α (TNFα), C-reactive protein (CRP), serum amyloid A (SAA), ECG, and HRV.	Exposure to PM_2.5_, UFP, noise, and gaseous co-pollutants was assessed during five consecutive work shifts. To control for post-work-shift exposure, personal PM_2.5_ in real time and noise exposure measurement was continued after the end of work (around 17:00 h) until the next morning.
5	Chung et al. [15]	Panel study	U.S., Somerville, Dorchester, South Boston	August 2009–June 2011	220 participants, (mean age = 58.5 years), white (66%; 68%), female (61%; 62%); >50% participants overweight (mean BMI 29.8; 29.6), 36% obese	Outdoor	Real environment	Monitoring station	UFP (size not defined)	Winter: 17 (5.8); summer: 8.3 (5.1)	BC and PM_2.5_	Continuous	Long-term outcome: SBP, DBP, and pulse pressure (PP)	In the morning, two visits in one year (winter and summer).
6	Kubesch et al. [16]	Cross-over real-world exposure study	Spain, Barcellona	February–November 2011	28 healthy non-smoking adults (15 female; 13 male); age range of 18–60 years; mean age 34.4	Outdoor	Real environment	Monitoring station	10–1000 nm	Low TRAP: 32.993 (12.422–56.735); High TRAP: 164.464 (80.346–344,297)	Black carbon (BC), fine particulate matter (PM_10_ and PM_coarse_), UFP, and nitric oxides (NO_x_).	2 h	Short-term outcome: SBP; DBP.	Pre, intra, exposure, 0.5 h, 2 h, 3 h, 4 h, and 5 h after exposure.
7	Pieters et al. [10]	Panel study	Belgium, Antwerp	spring (17 May–20 June) and fall (10 November–13 December) 2011	130 healthy children (6–12 years of age)	Outdoor	Real environment	Stationary particle counter	20–30, 30–50, 50–70, 70–100, 100–200, and >200	5.538 (25th percentile); 7.204 (75th percentile)	PM_2.5_, PM_10_, temperature, relative humidity, and wind speed.	Continuous	Long-term outcome: SBP, DBP; interleukin (IL)–1β (measured in Exhaled Breath Condensate)	Each child was examined twice in periods of about 26 weeks apart.
8	Padró-Martínez et al. [17]	Double-blind cross-over trial	Somerville, USA	February 2011–November 2012	20 participants (17 women); mean age 53.9	Indoor	Real environment	Stationary particle counter	7–3000	4.8	NA	Continuous	Long-term outcome: SBP, DBP, high sensitivity C-reactive protein (hsCRP), and interleukin-6 (IL-6), tumor necrosis factor alpha-receptor II (TNF-RII), fibrinogen, and IL-6.	On day 1, just before HEPA/sham filtration was started on day 21, 1–2 h before the filters were changed, and on day 42, just before the end of the intervention, for BP measures that lagged 0, 1 and 2 h.
9	Soppa et al. [18]	Cross-over sham-controlled exposure study	Germany, Düsseldorf,	NA	54 healthy adult men (27) and women (27); age 18–79 years (33 mean age); non-smoker or ex-smoker status for at least ten years	Indoor (candle burning (CB), toasting bread (T) and frying sausages (FS))	Laboratory	Monitoring chamber	<100 nm	Room air (baseline): 3; candle burning: 2670.0 ± 200.6; toasting bread: 1550.8 ± 170.6; Frying sausages: 600.7 ± 110.8	PM_1_ [μg/m³], LDSA [μm^2^/cm³], PM_10_ [μg/m³]	2 h	Short-term outcome: SBP; DBP.	Directly after, and 2 h, 4 h, and 24 h after exposure.
10	Chen et al. [19]	Panel study	China, Beijing	winter 2014	20 healthy, non-smoking male subjects (ages ranging 18–26 years)	Outdoor and indoor	Real environment	Fast mobility particle sizer	6–560	16.2 (7.3)	PM_2.5_, BC, accumulation mode particles (AMP)	Continuous	Long-term outcome: fractional exhaled nitric oxide (FeNO), cytokines in exhaled breath condensate, blood pressure, and pulse wave analysis (PWA).	Follow-up period at an outdoor fixed monitoring station beginning 14 days prior to each visit.
11	Corlin et al. [20]	Longitudinal	USA, Massachusetts	2004–2015	791 adults (69% women) participated in the longitudinal Boston Puerto Rican Health Study; mean age 57.	Outdoor and indoor	Real environment	Model accounting for spatial and temporal trends	<100	23 (3.4)		Continuous	Long-term outcome: SBP, DBP, hsCRP, particle inhalation rate (PIR).	Participants were visited up to three times over approximately six months. The mean time between 1st and 2nd visit was 2.2 years, while the mean time between the 2nd and 3rd visit was 4.1 years.
12	Liu et al. [21]	Panel study	Northern Taiwan	January 2014–August 2017	100 healthy adults (non-smoking, age range of 20–64 years); 50% women	Indoor	Real environment	Stationary particle counter	50–100	1.47 ± 0.88 [μg/m^3^]	PM_10_, PM_2.5_, NO_2_, O_3_, temperature and relative humidity.	Continuous	Long-term outcome: SBP, DBP, Forced Expiratory Volume in 1 s (FEV1), protein C (hs-CRP).	Each participant was interviewed and examined three times with a one-month break.
13	Schubauer-Berigan et al. [22]	Cross-sectional	US	December 2012–September 2014	108 workers from the carbon nanotubes industry	Indoor	Occupational environment	Personal monitoring	10–1000 nm	6.22 (41.2) [μg/m^3^]	NA	At least two days of full shift.	Short-term outcome: SBP, DBP, and percent predicted (PP) values for FVC, FEV1/FVC% (using the largest valid FEV1 and FVC), and FEF25-75%.	Blood samples were collected within hours.
14	Gabdrashova et al. [23]	Cross-sectional	Kazakhstan, Nur-Sultan	NA	Phase 1: 17 healthy non-smoking adults (7 men and 10 women, aged 18–46 years); Phase 2:33 non-smoking healthy adults (11 men and 21 women, aged 18–51 years)	Controlled laboratory (cooking beef)	Laboratory	Condensation Particle Counter	1–100	Phase 1: 5; Phase 2: 20	Indoor temperature, relative humidity (RH), CO_2_	Phase 1: study the effect of the exposure and post-exposure (up to 30 min) periods on HRV and BP. Phase 2: conducted with an extended post-exposure period, up to 120 min after the end of the cooking.	Short-term outcome: SBP, DBP, and HRV.	Before cooking, at the end of cooking, 60, 90, and 120 min after the cooking.
15	Lin et al. [24]	Cross-sectional	Northeast China	2006–2009	24 845 adults (aged 18–74 years; mean 43.7)	Outdoor and indoor	Real environment	Chemical transport model	<100	5.9 ± 0.8 μg m^−3^ with the range of 4.5–6.8 μg m^−3^	NA	Continues (>5 years living in the selected zone)	Long-term outcome: SBP, DBP, mean arterial pressure (MAP), pulse pressure (PP), prehypertension, and hypertension.	During follow-up (frequency NA).
16	Lyu et al. [25]	Cross-sectional	China, Beijing	NA	53 printing room workers and 54 controls, age > 16 years, mean age 31.5	Indoor (occupational)	Real environment	Stationary particle counter	20–850	workplace: 9.968 ± 4.665, control: 4.667 ± 1.840	O_3_ and volatile organic chemicals (VOCs)	Cumulative exposure time was 1173–132,860 h	Short-term outcome: DBP, SBP, MAP, Forced Vital Capacity (FVC), the percentage of the predicted FVC (FVC%), FEV1, FEV1%, peak expiratory flow (PEF), and PEF%.	NA
17	Guo et al. [26]	Panel study	Guangzhou, China	December 2017–January 2018	88 healthy university students	Outdoor	Real environment	Real-time sampling	<100	20.2 (11.7)	PM_2.5_, PM_1.0_, PM_0.5_, PM_0.2_, organic carbon (OC), elemental carbon (EC), and total carbon.	Continuous	Short-term effects: SBP and DBP.	Five weekly visits were taken for each participant over the entire study; Thursday/Friday from 6:00 to 10:00 p.m.
18	Roswall et al. [27]	Cohort	Denmark	2015–2019	32,851 adult Danes from the Diet, Cancer and Health—Next Generations cohort, mean age of 42.5	Outdoor and indoor	Real environment	AirGIS modeling system	NA	8.539 ± 2.354	PM_2.5_, UFP, EC and NO_2_, noise, intensity of traffic, meteorology, and street and building configurations; noise.	Continuous	Long-term outcome: high-density lipoprotein (HDL), non-high-density lipoprotein (non-HDL), SBP, and DBP.	Time windows from 24 h up to 90 days before blood sampling.
19	Gilbey et al. [28]	Cross-sectional	Perth, Western Australia	March 2017–May 2018	40 adults, healthy non-smokers aged between 35 and 69 years; mean age 52.6	Indoor	Real environment	Personal monitoring	<1000	11.256 (8744)	PM_2.5_, UFP.	Continuous, 24 h	Short-term outcomes: SBP, DBP, heart rate, augmentation index (%; AIx), augmented pressure (AP), pulse pressure (PP), and mean arterial pressure (MAP).	The mean BP and haemodynamic parameter measurements were calculated as the mean of all readings throughout the 24 h monitoring period.

## Data Availability

Not applicable.

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
