# Peer review of "Short- and Long-Term Effects of Inhaled Ultrafine Particles on Blood Pressure: A Systematic Review and Meta-Analysis"

_jcm, 2024, doi:10.3390/jcm13226802_

Round 1
Reviewer 1 Report
Comments and Suggestions for Authors
Review Summary of Manuscript: "Short-term and long-term effects of inhaled ultrafine particles on blood pressure: A systematic review and meta-analysis" by Lachowicz et al.
The manuscript investigates the effects of ultrafine particles (UFPs) on blood pressure, addressing both short-term and long-term consequences. The authors highlight that UFP exposure is a significant contributor to cardiovascular conditions, linked to a notable percentage of global premature mortality. Through systematic review and meta-analysis, the study reveals that short-term exposure to UFPs causes a temporary increase in blood pressure, with a more pronounced rise in systolic blood pressure (SBP) compared to diastolic blood pressure (DBP), returning to baseline within hours. Conversely, chronic exposure is associated with persistent SBP elevation and lower DBP, indicating a potential risk for worsening cardiovascular health, particularly among vulnerable populations in polluted areas. The study emphasizes the need for heightened awareness and interventions to address the health risks tied to UFP exposure.
Strengths:
· The manuscript is generally well-organized, presenting significant findings that highlight the health risks associated with UFP exposure.
Suggestions for Improvement:
1. Clarity and Organization:
o Condense the abstract to focus on key findings and implications, incorporating more quantitative data.
2. Introduction:
o Need to improve the research gap /study aims to address.
o Discuss limitations in existing studies on UFPs to provide context.
3. Methods:
o Elaborate on the rationale for excluding certain study designs.
o Provide additional details regarding the statistical analysis methods.
4. Results:
o Reasons for observed heterogeneity in findings should be clear.
5. Discussion:
o Relate findings to previous studies for a broader context.
o Emphasize public health implications, particularly for vulnerable populations.
6. Conclusions:
o Strengthen the conclusion by emphasizing the need for further research, including specific directions for future studies.
Final Recommendation: The manuscript presents significant findings that could have a substantial impact with minor revisions for clarity and presentation.
Comments on the Quality of English Language
can be improved
Author Response
Dear Editor and Reviewer,
First of all, together with all of the authors, I would like to thank you for your valuable revision and suggestions regarding our manuscript, to which we all agree. Please find in the following brief descriptions of the changes (marked in red in the revised manuscript) we made in order to improve our paper.
In below, we list the changes made according to Reviewer’s suggestions:
1. Clarity and Organization:
o Condense the abstract to focus on key findings and implications, incorporating more quantitative data.
Reply: The abstract section (page 1) was re-written according to Journal’s template. More quantitative data were included.
2. Introduction:
o Need to improve the research gap /study aims to address.
Reply: The research gap and study aims were improved (page 3).
•
Discuss limitations in existing studies on UFPs to provide context.
Reply: The limitations in existing studies on UFPs were provided (page 3, 25).
3. Methods:
o Elaborate on the rationale for excluding certain study designs.
Reply: The rationale for excluding certain studies was elaborated (page 4). Moreover, the main characteristics of studies included in the systematic review but not in the meta-analysis were presented in Table S4 (new table).
•
Provide additional details regarding the statistical analysis methods.
Reply: Additional details regarding the statistical analysis methods were provided (page 5-6; Text S2).
2
4. Results:
o Reasons for observed heterogeneity in findings should be clear.
Reply: The reasons for observed heterogeneity (I2 values provided in Figure 3) have been provided.
5. Discussion:
o Relate findings to previous studies for a broader context.
Reply: The findings to previous studies were related to a broader context (page 25-27).
•
Emphasize public health implications, particularly for vulnerable populations.
Reply: The public health implications, particularly for vulnerable populations were emphasized (page 25).
6. Conclusions:
o Strengthen the conclusion by emphasizing the need for further research, including specific directions for future studies.
Reply: The need for further research focusing on vulnerable populations was strengthened in the conclusions section.
Kind regards
Joanna I. Lachowicz

Reviewer 2 Report
Comments and Suggestions for Authors
I thank the publisher for giving me the opportunity to review this article. The article by Lachowicz and Gac is a systematic review of the literature on the role of UFPs on blood pressure. The authors have done a good and intense job, however I believe that some changes are necessary. 1. English editing is necessary, some sentences are incomprehensible. 2. Reduce plagiarism. 3. Incorrect abbreviations. 4. Quotes do not follow a single style. 5. Paragraph numbering is incorrect. Let's get to the structure of the text: Abstract: necessary organization in introduction, materials and methods.. Introduction: it is necessary to reduce the length. Materials and methods: sufficient. Results: sufficient. Tables: modify table1. Adjust quotes and caption. Discussion: Cite ESC 2024 guidelines on hypertension. Conclusions and limitations: sufficient.
Comments on the Quality of English LanguageEnglish editing is necessary, some sentences are incomprehensible.
Author Response
Dear Editor and Reviewer,
First of all, together with all of the authors, I would like to thank you for your valuable revision and suggestions regarding our manuscript, to which we all agree. Please find in the following brief descriptions of the changes (marked in red in the revised manuscript) we made in order to improve our paper.
In below, we list the changes made according to Reviewer’s suggestions:
1.
English editing is necessary, some sentences are incomprehensible.
Reply: The revised manuscript has been edited by the professional English Editor.
2.
Reduce plagiarism.
Reply: Plagiarism has been reduced.
3.
Incorrect abbreviations.
Reply: The abbreviations has been revised
4.
Quotes do not follow a single style.
Reply: The style of the quotes has been standardized.
5.
Paragraph numbering is incorrect.
Reply: The paragraph numbering has been corrected.
Let's get to the structure of the text:
Abstract: necessary organization in introduction, materials and methods.
Reply: The abstract section has been re-written.
Introduction: it is necessary to reduce the length.
Reply: We were unable to reduce the length without interfering with other reviewers comments and suggestion.
Tables: modify table1. Adjust quotes and caption.
Reply: Quotes and captions have been revised.
2
Discussion: Cite ESC 2024 guidelines on hypertension.
Reply: ECS 2024 guidelines on hypertension has been referenced.
Kind regards
Joanna I. Lachowicz

Reviewer 3 Report
Comments and Suggestions for Authors
Thank you for inviting me to peer review this systematic review and meta-analysis on the effects of ultrafine particle (UFP) exposure on blood pressure. This is an important and timely study. The authors analyzed the overall impact of UFP exposure on systolic blood pressure (SBP) and diastolic blood pressure (DBP) and, for the first time, focused on short-term changes in the SBP/DBP ratio (although specific results were not presented). This study provides valuable insights into understanding the cardiovascular health effects of UFPs. Below are my detailed review comments:
1. Heterogeneity is a common challenge in meta-analyses, especially in complex environmental exposure studies. In this study, heterogeneity is mainly manifested in the following aspects:
- Study design heterogeneity: The 13 included studies encompass various types of study designs, such as cross-sectional studies, cohort studies, panel studies, and crossover designs. Some studies focus on short-term effects (6 studies), while others focus on long-term effects (7 studies). Study durations range from hours to years, making direct comparisons of results difficult.
- Exposure assessment method heterogeneity: These include personal exposure assessments, fixed monitoring stations, air quality models, etc. Some studies were conducted under laboratory conditions (e.g., candle burning and beef cooking), while others were conducted in real-world environments.
- The study did not clearly specify the age range of participants in the included studies, which may lead to heterogeneity between different age groups.
- It needed to be clearly stated whether only healthy populations were included or if patients with specific diseases were also included.
- Outcome measurement heterogeneity: The time points and frequency of blood pressure measurements varied across studies. Some studies measured blood pressure only once, while others conducted multiple measurements or 24-hour ambulatory blood pressure monitoring.
- Influence of air pollutants: Some studies simultaneously measured other air pollutants (such as PM2.5, NO2, etc.), while others focused only on UFPs, which may affect the estimation of independent effects of UFPs.
The authors need to explain these heterogeneities in detail.
2. For handling heterogeneity, it is recommended to use quantitative methods such as the I² statistic and Cochran's Q test to assess the degree of heterogeneity. It is suggested that the authors consider conducting meta-regression analysis to explore potential factors causing heterogeneity. Alternatively, more detailed subgroup analyses could be performed, such as those based on population characteristics or exposure assessment methods.
3. For publication bias assessment, it is recommended that the authors provide more objective statistical tests, such as Egger's test or Begg's test. While funnel plots were presented in the supplementary materials of the article, it is advised to describe their structure and interpretation in detail.
4. Although the authors provided keywords in the article, they did not provide a complete search strategy, including all restrictions and database-specific syntax. It is recommended that the authors provide a complete search strategy, including all restrictions and database-specific syntax, preferably as supplementary material. The authors searched two major medical literature databases, PubMed and Scopus. The reviewer is concerned about insufficient database coverage and suggests supplementing the search with other important databases such as Web of Science and Embase. Additionally, considering the potential time gap between article completion or submission and review (From October 2023 to October 2024), it is recommended to conduct an updated search to ensure the inclusion of the latest research.
5. Regarding inclusion and exclusion criteria, the study did not clearly specify the age range of participants in the included studies, which may lead to heterogeneity between different age groups. Furthermore, the article did not clearly state whether only healthy populations were included or if patients with specific diseases were also included.
6. The study did not clearly state whether a fixed-effects model or a random-effects model was used. Given the potential heterogeneity between studies, a random-effects model might be more appropriate, but this point was not discussed. Additionally, while the study focused on UFP exposure, it did not explore the relationship between exposure levels and effect sizes. If possible, a dose-response relationship analysis should be conducted to explore the relationship between UFP exposure levels and blood pressure changes. Alternatively, consider using more advanced statistical methods, such as network meta-analysis, to compare multiple exposure levels or different types of particles simultaneously.
Author Response
Dear Editor and Reviewer,
First of all, together with all of the authors, I would like to thank you for your valuable revision and suggestions regarding our manuscript, to which we all agree. Please find in the following brief descriptions of the changes (marked in red in the revised manuscript) we made in order to improve our paper.
In below, we list the changes made according to Reviewer’s suggestions:
1. Heterogeneity is a common challenge in meta-analyses, especially in complex environmental exposure studies. In this study, heterogeneity is mainly manifested in the following aspects:
- Study design heterogeneity: The 13 included studies encompass various types of study designs, such as cross-sectional studies, cohort studies, panel studies, and crossover designs. Some studies focus on short-term effects (6 studies), while others focus on long-term effects (7 studies). Study durations range from hours to years, making direct comparisons of results difficult.
Reply: The reasons for heterogeneity have been described wider (page 22).
- Exposure assessment method heterogeneity: These include personal exposure assessments, fixed monitoring stations, air quality models, etc. Some studies were conducted under laboratory conditions (e.g., candle burning and beef cooking), while others were conducted in real-world environments.
Reply: The reasons for heterogeneity have been described wider (page 22).
- The study did not clearly specify the age range of participants in the included studies, which may lead to heterogeneity between different age groups.
Reply: The reason for heterogeneity has been described wider (page 22). The age range and mean age of the participants of the selected studies are provided in Table 1.
- It needed to be clearly stated whether only healthy populations were included or if patients with specific diseases were also included.
2
Reply: The health status of the participants of the selected studies has been defined in Table 1 and in the main text (section 3.4. Results of individual studies).
- Outcome measurement heterogeneity: The time points and frequency of blood pressure measurements varied across studies. Some studies measured blood pressure only once, while others conducted multiple measurements or 24-hour ambulatory blood pressure monitoring.
Reply: The reasons for heterogeneity (among which time of BP measure after exposure) has been described wider (page 22).
- Influence of air pollutants: Some studies simultaneously measured other air pollutants (such as PM2.5, NO2, etc.), while others focused only on UFPs, which may affect the estimation of independent effects of UFPs.
The authors need to explain these heterogeneities in detail.
Reply: The reasons for heterogeneity (among which other air pollutants) has been described wider (page 22).
2. For handling heterogeneity, it is recommended to use quantitative methods such as the I² statistic and Cochran's Q test to assess the degree of heterogeneity. It is suggested that the authors consider conducting meta-regression analysis to explore potential factors causing heterogeneity.
Reply: The I2 values are provided in Figure 3.
Alternatively, more detailed subgroup analyses could be performed, such as those based on population characteristics or exposure assessment methods.
Reply: We are unable to perform good quality subgroup analyses due to the low number of studies in each subgroup.
3. For publication bias assessment, it is recommended that the authors provide more objective statistical tests, such as Egger's test or Begg's test. While funnel plots were presented in the supplementary materials of the article, it is advised to describe their structure and interpretation in detail.
Reply: We were unable to prepare for Egger’s and/or Begg’s test while the number of analyzed studies is below 10.
4. Although the authors provided keywords in the article, they did not provide a complete search strategy, including all restrictions and database-specific syntax. It is recommended that the authors provide a complete search strategy, including all restrictions and database-specific syntax, preferably as supplementary material.
Reply: The search strategy has been revised (page 4 and Supplementary Material file).
The authors searched two major medical literature databases, PubMed and Scopus. The reviewer is concerned about insufficient database coverage and suggests supplementing the search with other important databases such as Web of Science and Embase. Additionally, considering the potential time gap between article completion or submission and review (From October 2023 to October
3
2024), it is recommended to conduct an updated search to ensure the inclusion of the latest research.
Reply: The search study has been repeated with the longer times range (1stJanuary 2013 – 9th October 2024) and was executed in 4 databases: Embase, WebOfScience, Scopus and PubMed. New data has been integrated into the Systematic Review.
5. Regarding inclusion and exclusion criteria, the study did not clearly specify the age range of participants in the included studies, which may lead to heterogeneity between different age groups. Furthermore, the article did not clearly state whether only healthy populations were included or if patients with specific diseases were also included.
Reply: the age of participants and health status in each study was reported in Table 1.
6. The study did not clearly state whether a fixed-effects model or a random-effects model was used.
Reply: The results of fixed- and random-effects models are presented in Figure 3.
Given the potential heterogeneity between studies, a random-effects model might be more appropriate, but this point was not discussed. Additionally, while the study focused on UFP exposure, it did not explore the relationship between exposure levels and effect sizes. If possible, a dose-response relationship analysis should be conducted to explore the relationship between UFP exposure levels and blood pressure changes. Alternatively, consider using more advanced statistical methods, such as network meta-analysis, to compare multiple exposure levels or different types of particles simultaneously.
Reply: Unfortunate, we are not able to deliver the good quality dose-response relationship analysis due to insufficient number of studies with proper data for such analysis.
Kind regards
Joanna I. Lachowicz

Round 2
Reviewer 2 Report
Comments and Suggestions for Authors
Author have modified the text following requests.
Comments on the Quality of English LanguageMinor editing
Author Response
Dear Editor and Reviewer,
Together with all of the authors, I would like to thank you for your time dedicated to the revision process.
In below, we list the change made according to Reviewer’s suggestions:
Comments on the Quality of English Language: Minor editing
Reply: We run the minor English Editing throughout the main text.
Kind regards
Joanna I. Lachowicz

Reviewer 3 Report
Comments and Suggestions for Authors
Thank you to the authors for thoughtfully responding to the reviewer's comments. The quality of the manuscript has improved after the revisions, but some areas still need further improvement, particularly in the analysis of dose-response relationships and handling of heterogeneity. The reviewer believes that further improving these aspects will enhance the scientific rigor and impact of the paper.
1. The authors explained that a dose-response analysis could not be conducted due to insufficient data. However, this remains a significant limitation. Dose-response relationships are especially important in environmental exposure studies, as they help establish a direct link between exposure levels and health outcomes. The authors could further discuss this point and suggest how future studies should be designed to fill this gap. Even if the current data are insufficient, the authors could explore potential dose-response patterns through sensitivity analysis or other methods (it is recommended to consult a biostatistician if necessary). If the number of studies is too limited to conduct any meaningful sensitivity analysis, the authors should clearly explain this limitation in the manuscript and provide a rationale for why sensitivity analysis could not be performed. This transparent reporting will help readers understand the limitations of the study.
2. The authors explained the sources of heterogeneity and provided the I² statistic, but heterogeneity remains a major issue due to the diversity of the studies. Since the small number of studies prevented subgroup or meta-regression analyses, the study's overall conclusions may be influenced by heterogeneity. The authors are advised to further emphasize how heterogeneity affects the interpretation of the results in the discussion section and suggest that future studies could reduce heterogeneity by standardizing study designs and exposure assessment methods.
3. The authors distinguished between the effects of short-term and long-term exposures, but the definitions and explanations of these two types of exposure may be overly broad. For example, "short-term" exposure is defined as less than 7 days, while "long-term" exposure is defined as more than 7 days. This time classification may be too simplistic to capture the differences in health effects across different time scales. The author is encouraged further to refine the definitions of short-term and long-term exposures, or at least acknowledge the limitations of this distinction in the discussion, and explore the potential differences in health effects across different time scales.
4. The conclusion section is somewhat lengthy and repeats some of the content from the discussion. While it is important to summarize the findings, too much detail in the conclusion may make it difficult for readers to grasp the key points. The authors are advised to simplify the conclusion section, highlighting the most important findings and recommendations for future research while leaving more detailed discussions in the discussion section.
Author Response
Dear Editor and Reviewer,
Together with all authors, I would like to thank you for your time dedicated to the revision process. We would like to thank you also for your latest revision and suggestions regarding our manuscript, to which we all agree. Please find in the following brief descriptions of the changes (marked in red in the revised manuscript) we made in order to improve our paper.
In below, we list the changes made according to Reviewer’s suggestions:
- The authors explained that a dose-response analysis could not be conducted due to insufficient data. However, this remains a significant limitation. Dose-response relationships are especially important in environmental exposure studies, as they help establish a direct link between exposure levels and health outcomes. The authors could further discuss this point and suggest how future studies should be designed to fill this gap. Even if the current data are insufficient, the authors could explore potential dose-response patterns through sensitivity analysis or other methods (it is recommended to consult a biostatistician if necessary). If the number of studies is too limited to conduct any meaningful sensitivity analysis, the authors should clearly explain this limitation in the manuscript and provide a rationale for why sensitivity analysis could not be performed. This transparent reporting will help readers understand the limitations of the study.
Reply: The reason for the lack of dose-response analysis and potential effect on heterogeneity was emphasized in 2.6. Sensitivity analysis section (page 5 and 6), and in 3. Results section (page 8, 23),
- The authors explained the sources of heterogeneity and provided the I² statistic, but heterogeneity remains a major issue due to the diversity of the studies. Since the small number of studies prevented subgroup or meta-regression analyses, the study's overall conclusions may be influenced by heterogeneity. The authors are advised to further emphasize how heterogeneity affects the interpretation of the results in the discussion section and suggest that future studies could reduce heterogeneity by standardizing study designs and exposure assessment methods.
Reply: We emphasized the fact that missing dose-response analysis due to low number of studies and due to lack of studies focusing on dose-response and time-dose-effects influenced heterogeneity. Also, we suggested that future studies analyzing both time- and dose-response axes could reduce the existing gap (page 26).
- The authors distinguished between the effects of short-term and long-term exposures, but the definitions and explanations of these two types of exposure may be overly broad. For example, "short-term" exposure is defined as less than 7 days, while "long-term" exposure is defined as more than 7 days. This time classification may be too simplistic to capture the differences in health effects across different time scales. The author is encouraged further to refine the definitions of short-term and long-term exposures, or at least acknowledge the limitations of this distinction in the discussion, and explore the potential differences in health effects across different time scales.
Reply: The 7-days cut-off between short- term and long-term effects of air pollutants on human health is classical distinction in the studies of air pollutants and their effects on human health. We acknowledged the limitations of this distinction in the discussion section (page 27).
- The conclusion section is somewhat lengthy and repeats some of the content from the discussion. While it is important to summarize the findings, too much detail in the conclusion may make it difficult for readers to grasp the key points. The authors are advised to simplify the conclusion section, highlighting the most important findings and recommendations for future research while leaving more detailed discussions in the discussion section.
Reply: The conclusion section was significantly shortened by removing the lengthy repeats of discussion. We have also highlighted the need for future research.
Kind regards
Joanna I. Lachowicz
